# Carbon defect qubit in two-dimensional WS$_2$

Song Li[1], Gergő Thiering[1], Péter Udvarhelyi [1], Viktor Ivády[1,2,3] & Adam Gali [1,4 ✉]

Identifying and fabricating defect qubits in two-dimensional semiconductors are of great interest in exploring candidates for quantum information and sensing applications. A milestone has been recently achieved by demonstrating that single defect, a carbon atom substituting sulphur atom in single layer tungsten disulphide, can be engineered on demand at atomic size level precision, which holds a promise for a scalable and addressable unit. It is an immediate quest to reveal its potential as a qubit. To this end, we determine its electronic structure and optical properties from first principles. We identify the fingerprint of the neutral charge state of the defect in the scanning tunnelling spectrum. In the neutral defect, the giant spin-orbit coupling mixes the singlet and triplet excited states with resulting in phosphorescence at the telecom band that can be used to read out the spin state, and coherent driving with microwave excitation is also viable. Our results establish a scalable qubit in a two-dimensional material with spin-photon interface at the telecom wavelength region.

[1] Wigner Research Centre for Physics, P.O. Box 49, Budapest H-1525, Hungary. [2] Department of Physics, Chemistry and Biology, Linköping University, 581 83 Linköping, Sweden. [3] Max Planck Institute for the Physics of Complex Systems, Nöthnitzer Straße 38, 01187 Dresden, Germany. [4] Department of Atomic Physics, Institute of Physics, Budapest University of Technology and Economics, Műegyetem rakpart 3., H-1111 Budapest, Hungary. ✉email: gali.adam@wigner.hu

Point defects with spins in solids which constitute a two-level system are considered as essential building blocks for application in quantum information, computing and sensing[1–4] when the electron spin can be initialized and read out with sufficiently long coherence time. The nitrogen-vacancy (NV) centre in diamond has been already identified as a qubit[5–8] that can be exploited in diverse quantum technology applications[1–4,9]. The NV qubit state is initialised and read out by optical means in which spin-photon interface can be realized for quantum communication in the visible wavelength region[10]. However, NV centre in diamond is relatively dim, and scalable fabrication and optical waveguide integration of NV centre in diamond and akin defect qubits in three-dimensional (3D) crystals is technologically very challenging[11–13].

Compared to single photon emitters (SPE) and qubits in 3D materials, defects in two-dimensional (2D) wide bandgap host distinguish themselves with high quantum efficiency and additional functionalities due to spatial confinement and near-surface location[14,15]. Recently, coherent manipulation of single spins have been reported in van der Waals 2D materials by means of optical initialisation and read-out[16,17]. In particular, carbon-related defects have been associated with these qubits in hexagonal boron-nitride (hBN)[17–19] but scalable, deterministic preparation or activation of the qubit defects has not yet been achieved. Although, ab initio calculations proposed the atomic composition of the qubits[17,18] but it has not yet been unambiguously confirmed which stems to create them on demand. Alternatively, boron-vacancy, an identified defect, in hBN may act as qubit[17,20–22]. Boron-vacancies are created by irradiation[17] or other invasive techniques[23] but no deterministic single defect creation is in reach and no single spin boron-vacancy qubit has been reported to date. Nevertheless, these efforts motivate scientists to design and fabricate quantum defects in other 2D materials.

Transition metal dichalcogenides (TMDCs) is another ideal 2D host for quantum defects[24–26]. Single photon emitters have been measured in WSe$_2$[24,27,28] and WS$_2$, but no coherent control of single spins have been reported[24,29]. The inherit spin-orbit-coupling (SOC) and excitonic effect provides long coherence time and electric controlled spin-valley coupling[30,31]. Recent investigation realized the creation of single carbon defect in WS$_2$ through scanning tunnelling microscopy (STM) tip with atomic precision[32], in which carbon replaces sulphur, that could isolate a single defect with a doublet spin state. The highly deterministic

location with charge and spin state control provides large-scale defect arrays for integrated quantum systems. Furthermore, the electronic structure of the prepared single defect could be directly measured through scanning tunnelling spectra (STS) in TMDCs. However, it has not been confirmed that the single carbon defect spin can be exploited as a qubit in WS$_2$, i.e., no quantum protocol to initialise and read out the spin state and means to coherently drive the spin states have been demonstrated.

In this paper, we demonstrate that single carbon defect spin in WS$_2$, which can be deterministically prepared at atomic scale precision, can act as a qubit where the spin-photon interface can be realized with emission in the telecom wavelength region. We first carry out ab initio magneto-optical spectroscopy to explore the quantum states generated by the carbon defect. We identify the triplet state of the neutral carbon defect (C$_S$) acting as a suitable qubit state with telecom wavelength phosphorescence. The multiplets of the triplet state can be used as a resource to prepare a given quantum state and store quantum information. We show a quantum optics protocol to initialise, read out and drive the single spin of C$_S$ defect in WS$_2$, which converts a scalable defect spin to a promising qubit in a van der Waals material.

## Results and discussion

**Ground state calculations.** Figure 1a shows the considered C$_S$ defect. Carbon replaces sulphur atom on the top layer. This could be formed by hydrogen depassivation of carbon-hydrogen impurity through voltage pulse from the STM tip. The graphene substrate adjusted the Fermi-level alignment, and experiment confirmed the defect exhibit negatively charged state as C$_S^-$[32]. One occupied state and unoccupied state were detected at $-0.4$ V and $+0.52$ V sample bias, respectively. We could identify the two defect levels of C$_S^-$ in the ground state by density functional theory (DFT) HSE calculation (see Methods). We note that charge correction is needed to improve the DFT total energy and defect level, as discussed in the Supplementary Note 3. In addition, spin-orbit coupling (SOC) significantly contributes to the electronic structure. For the pristine WS$_2$, the position of valence band maximum (VBM) shifts with respect to the value without SOC, and the final calculated band gap is 2.76 eV. Another consequence of SOC is that the double degenerate VBM splits. Similarly, SOC modifies the defect levels as indicated in Fig. 1c. The calculated occupied defect level lies at 1.09 eV above VBM. The STS measured occupied level is roughly about 1.1 eV above the VBM considering the spectra smearing (about 0.19 eV)[32], in good agreement with the theoretical value. In addition to the in-gap defect levels,

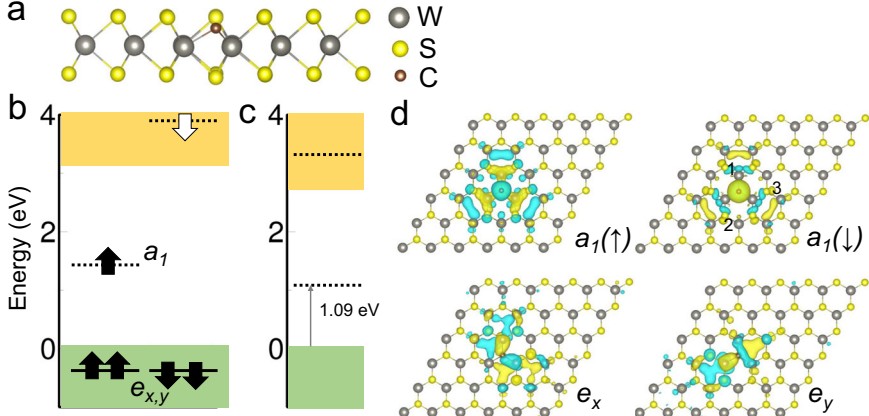

**Fig. 1 Electronic structure of the C$_S^-$ in WS$_2$. a** Side view of the optimized ground state geometry of C$_S^-$. The energy levels diagram of C$_S^-$ at ground state without (**b**) and with (**c**) SOC effect. The arrows indicate spin-up and spin-down channels. The dash lines indicate the position of defect levels after charge correction. **d** Spatial wave function of the defect levels. Numbers denote the first, second and third neighbouring tungsten atoms around carbon atom. The isosurface is set to 0.00068/Å³.

there are degenerate $e$ defect levels ($e_x$, $e_y$) about $-0.62$ eV below the VBM. The degeneracy is lifted due to SOC interaction and the splitting is 113 meV. Such large splitting has been previously measured for sulphur vacancy in $WS_2$ by STS[31]. Figure 1d plots the spatial wavefunction of the defect levels. The distribution of in-gap levels clearly demonstrates $C_{3v}$ symmetry with $a_1$ character. The degenerate states mainly come from the $d$ orbitals of neighbour tungsten atoms. However, the unoccupied defect level is located far above the conduction band minimum. Hence the experimentally observed occupied defect level is attributed to the negatively charged defect while the unoccupied level does not correspond to $C_S^-$.

We anticipate that, with positive bias in STS, the Fermi-level of the sample shifts down with respect to the Fermi-level of the tip. As a consequence, the electron from the occupied defect level in the gap transfers to the graphene substrate, so the defect level is depleted in the gap with turning the defect charge state from negative to neutral. With the increase of sample bias to +0.52 eV, electron from the tip transfers to the unoccupied state of the neutral $C_S^0$ and recharges it. Such charge transition during STS measurement has been reported for the silicon dangling bond on the Si surface[33]. Hence, the unoccupied state observed with positive bias should be associated with the empty level of the neutral defect. This scenario is confirmed by our calculations. The calculated formation energy as a function of Fermi-level is plotted in Fig. 2. The charge transition level (CTL) for $(0|-1)$ is at 1.57 eV above VBM. The graphene substrate tunes this Fermi level of $WS_2$ above CTL and maintains negatively charged. However, positive bias sinks the Fermi-level and alters the charge state during experiment. The defect levels of neutral $C_S^0$ is plotted at Fig. 2b. There is only one non-degenerate unoccupied $a_1$ level in the gap at 2.11 eV above VBM which again agrees with the observed one at about 2.1 eV above VBM in STS[32].

Our interpretation of results clearly shows that the charge state of single $C_S$ defect can be well manipulated by the choice of the substrate and/or the applied bias. The spin state of $C_S^-$ is $S = 1/2$ which might act as qubit. Nevertheless, the calculated excitation energies with involving the defect state in the gap are always larger than the ionisation threshold energy at around 1.2 eV (see Supplementary Note 4). This makes the optical readout of the spin state difficult as ionisation competes with the neutral photo-excitation of the defect upon illumination. The $C_S^0$ defect ground state is a closed shell singlet but a triplet $S = 1$ metastable state may exist that can act as a qubit similarly to the ST1 centre in diamond[34]. Next, we explore the fine electronic structure of $C_S^0$ in $WS_2$ and its potential as qubit.

**Ab initio spectroscopy of $C_S^0$ in $WS_2$.** In the neutral charge state, there is one empty $a_1$ level in the gap. The ground state is a closed shell singlet $^1A_1$. An electron from VBM may be promoted to the empty $a_1$ level upon illumination. After emptying the VBM state, a localized $e$ level pops up in the gap because of the strong Coulomb interaction between the localized electrons which lies at 0.6 eV below VBM when occupied. In the hole picture, the excited state can be described as $ea_1$ electronic configuration and the corresponding many-body states can be analysed by group theory within $C_{3v}$ symmetry (e.g., Ref. [35]). One singlet, $^1E$ state exists near the $^3E$ triplet state. The $^3E$ manifold contains $E_{x, y}$ $m_s = 0$ and $A_1 \oplus A_2 = A_{1,2}$ and $E_{1,2}$ $m_s = \pm1$ spin projections in double group notation.

First, we approximate the singlet and triplet excitation energies by $e^\uparrow a_1^\downarrow$ and $e^\uparrow a_1^\uparrow$ electronic configurations (arrows represent the respective spin-up and spin-down spin channels), respectively, in $C_{3v}$ symmetry without SOC. The calculated ZPL energies are 1.062 eV and 0.985 eV, respectively. These energies are significantly smaller than the photo-ionisation threshold energy, thus $C_S^0$ is a photostable quantum emitter in the telecom wavelength region. Furthermore, a high spin triplet state exists that may be used as a qubit.

The fine structure of the excited states of $C_S^0$ depends on the complex interaction of phonons, orbitals and spins. Orbitals and spins are coupled by spin-orbit interaction whereas $^1E$ and $^3E$ states are Jahn-Teller active, i.e, strong electron-phonon coupling is expected by the symmetry distorting $E$ phonons, i.e., it establishes an $E \otimes e$ dynamic Jahn-Teller (DJT) system. Since the estimated energy gap between the singlet and triplet levels is $\Delta = 54.9$ meV and the calculated spin-orbit coupling connecting the $E_{\{x,y\}}$ and $^1E$ states is $\lambda_z = -54$ meV the singlet and triplet states may be coupled as

$$\hat{H}_e = \hat{W}_{ee} + \hat{H}_{SOC} = 0|^3E\rangle\langle^3E| + \Delta|^1E\rangle\langle^1E| + \lambda_z|^3E\rangle\langle^1E| = 0|E_{x,y}\rangle$$
$$\langle E_{x,y}| + \Delta|^1E\rangle\langle^1E| + \lambda_z|E_{1,2}\rangle\langle E_{1,2}| - \lambda_z|A_{1,2}\rangle\langle A_{1,2}| + \lambda_z|E_{x,y}\rangle\langle^1E|, \quad (1)$$

where the reference zero energy is the level of the triplet state before applying spin-orbit interaction. The negative sign of $\lambda_z$ is due to the hole left on the $e$ orbital. Eq. (1) clearly shows that the singlet and the $m_s = 0$ part of the triplet are mixed by spin-orbit interaction. The eigenvalues of the matrix provide the electronic part of SOC are discussed in Supplementary Note 5.

However, the strength of the effective spin-orbit coupling which separates the $A_{1,2}$, $E_{x,y}$ and $E_{1,2}$ levels can be significantly reduced due to the strong electron-phonon coupling which mixes the $m_l = +1$ and $m_l = -1$ components of the electronic wavefunctions known as Ham reduction[36]. In the $E \otimes e$ DJT system[37], the electron-phonon coupling may be described as

$$\hat{H}_{JT} = \hbar\omega_e(a_x^\dagger a_x + a_y^\dagger a_y + 1) + F(\hat{\sigma}_z\hat{x} + \hat{\sigma}_x\hat{y}), \quad (2)$$

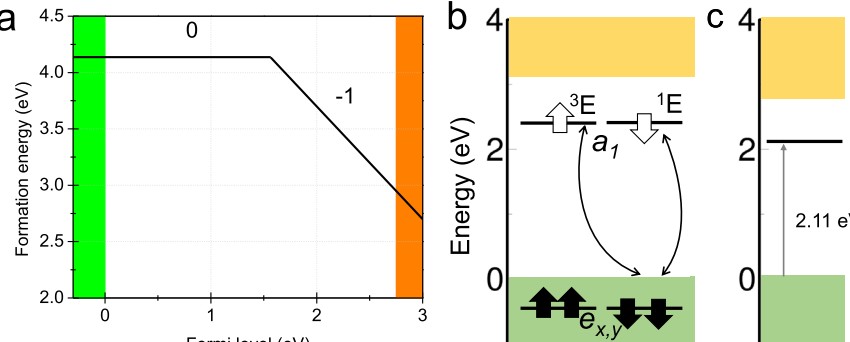

**Fig. 2 Thermal stability and electronic structure of the $C_S^0$ in $WS_2$. a** Formation energy as a function of Fermi-level for $C_S$ defect. The green and orange colours denote the VBM and CBM. The CTL energy is aligned to VBM with SOC included. The $-2$ charge state is not stable. The chemical potential of carbon atom is set to bulk diamond. The energy diagram of neutral state of $C_S$ at ground state without (**b**) and with (**c**) SOC effect.

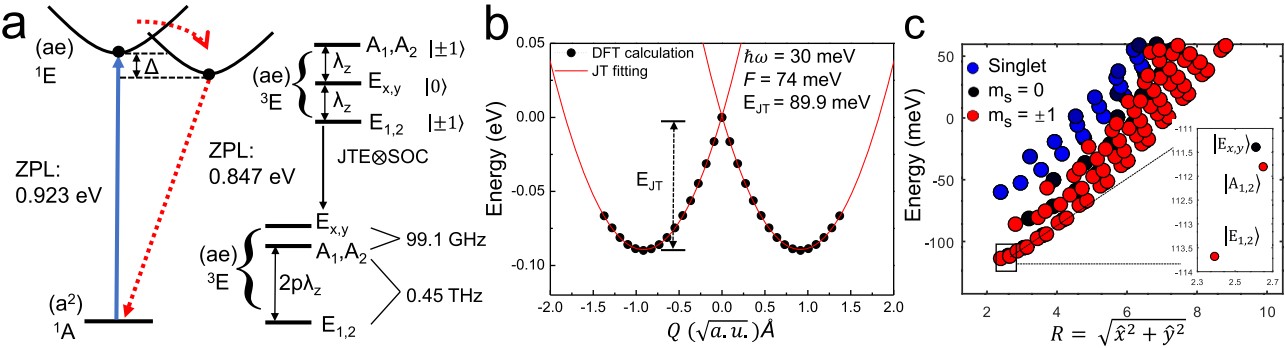

**Fig. 3 Multiplets structure and the JT process. a** The possible optical excitation cycle among ground state and excited states. Blue arrow indicates the allowed optical transition by electric dipole moment interaction. The black dots denote the structures with $C_s$ symmetry. The $E_{JT}$ are 139 meV and 89.9 meV for $^1E$ and $^3E$, respectively. Red dash lines indicate selective non-radiative ISC between different spin multiplicity. Intrinsic SOC splitting $\lambda_z$ separates the spin sublevels in the triplet $^3E$ excited state which is reduced by $p < 1$ Ham reduction factor due to Jahn-Teller (JT) interaction. **b** APES of the JTE between $C_{3v}$ and $C_s$ symmetry for $C_S^0$. The line is the fitted curve and dot is the DFT result. The $C_{3v}$ configuration is refered to $Q = 0$. The standard deviation is less than 1%. **c** Polaronic eigenstates for energy levels in $^3E$ with SOC $\otimes$ DJT interaction included. The inset shows the lowest vibronic solution for the three states.

where $a_{x,y}$, $a_{x,y}^\dagger$ are creation and annihilation operators for $E$ phonon, $\hbar\omega_e$ is the energy of the effective $E$ phonon mode. The $\hat{\sigma}_z = |e_x\rangle\langle e_x| - |e_y\rangle\langle e_y|$ and $\hat{\sigma}_x = |e_x\rangle\langle e_y| + |e_y\rangle\langle e_x|$. $F$ is the electron-phonon coupling strength and could be calculated from $F = \sqrt{2E_{JT}\hbar\omega_e}$. The $\hbar\omega_e$ can be obtained from parabolic fit to the adiabatic potential energy surface (APES) and $E_{JT} = 89.9$ meV is the JT energy as shown in Fig. 3(b). $E_{1,2}$ and $A_{1,2}$ cannot interact with $E_{x,y}$ and $^1E$ through DJT, and DJT acts only on the $m_s = \pm 1$ states. The calculated $F = 74$ meV is the same order of magnitude as the spin-orbit coupling strength. In this case, the spin-orbit interaction is not taken like a small perturbation near electron-phonon coupling but the full Hamiltonian $\hat{H} = \hat{W}_{ee} + \hat{H}_{SOC} + \hat{H}_{JT}$ should be solved simultaneously. Nevertheless, we solved this problem perturbatively too, in order to gain more insight about the complex interplay of phonons, orbitals and spins (see Supplementary Note 5). We find that the perturbative solution is surprisingly close to that of the full Hamiltonian solution, and the results can be explained by a Ham reduction factor $p = 0.017$ which means that the spin-orbit coupling is strongly quenched by the electron-phonon coupling with reducing the energy spacing between the $A_{1,2}$ and $E_{1,2}$ to about 454 GHz whereas the $E_{x,y}$ level goes above that of $A_{1,2}$ by about 99.1 GHz. As a consequence, the electron spin-spin interaction, typically in the GHz energy region, should be also considered in the spin Hamiltonian.

The electron spin-spin Hamiltonian of the $^3E$ state can be expressed by

$$\hat{H}_{ss} = D(|A_1\rangle\langle A_1| + |A_2\rangle\langle A_2| + |E_1\rangle\langle E_1| + |E_2\rangle\langle E_2|)$$
$$- 2D(|E_x\rangle\langle E_x| + |E_y\rangle\langle E_y|) \qquad (3)$$
$$+ 2\Lambda(|A_1\rangle\langle A_1| - |A_2\rangle\langle A_2|),$$

where the energy gaps between the $m_s = 0$ and $\pm 1$ and between the $A_1$ and $A_2$ are given by

$$D = \frac{\mu_0}{4\pi}g^2\beta^2\langle X|\frac{1 - 3\hat{z}^2}{4r^3}|X\rangle = -\frac{1}{2}D_{zz}, \qquad (4)$$

$$\Lambda = 2\frac{\mu_0}{4\pi}g^2\beta^2\langle X|\frac{3\hat{x}^2 - 3\hat{y}^2}{4r^3}|X\rangle = \frac{1}{2}D_{xx-yy}. \qquad (5)$$

Here, the $\beta$ is the Bohr magneton and $g$ is the Landé factor. The calculated $D$ is 2.5 GHz and $\Lambda$ is 970 MHz. The magnitude manifests itself as perturbation over SOC and DJT and could be treated independently. Due to the non-symmetric character of $\Lambda$,

it is reduced by the Ham factor discussed above and the reduced gap $p\Lambda$ is about 16.5 MHz. We note that the energy gap between $E_{x,y}$ and $A_1 \oplus A_2$ levels are indicative as will be discussed below.

The $E_{x,y}$ level shifts above the $m_s = \pm 1$ levels because its interaction with the singlet $^1E$ state. This interaction results in a singlet inmixture into the $E_{x,y}$ state by 2.1%. As a consequence, $C_S^0$ may act like a phosphorescence centre with emission between the triplet state and the singlet ground state. The calculated ZPL energy is 0.847 eV (~1460 nm) which nicely falls to the desired telecom wavelength for efficient quantum communication via optical fibre links. We note here that the estimated accuracy in the absolute value of the ZPL energy is about 0.1 eV (see Methods). Next, we discuss whether the triplet spin levels may be exploited to realize a qubit.

**Quantum optics protocol for $C_S^0$.** The intriguing electronic and phonon properties of $C_S^0$ motivate us to propose a set of quantum protocol for qubit application. We start at the $^1A_1$ ground state of the defect. The absorption by $^1E$ state upon illumination in the near infrared (NIR) region (about 1340 nm) should be very effective by $\{x, y\}$ polarized photons as the calculated optical transition dipole moment is about 24.7 Debye which corresponds to a radiative lifetime at 1.52 ns.

After arrival at $^1E$, a very fast intersystem crossing (ISC) occurs selectively to the $m_s = 0$ of $E_{x,y}$ state because of the following arguments. According to the theory and measurements for NV centre in diamond with similar electronic structure in the excited state,[38] the cooling of excited vibronic levels to the ground state vibronic level occurs within the order of 100 fs it is slowed down for $C_S^0$ to about 38.2 ps because of the overlap spin function of $^1E$ and $E_{x,y}$ state. Here we estimate the ISC rate from $^1E$ with

$$\tau_{ISC} = (2\pi/\hbar)\lambda_z^2 F(\Delta E), \qquad (6)$$

where $F$ is the function of overlap between the vibrational spectrum of singlet and triplet and $\Delta E$ is the energy splitting between singlet and triplet. The final estimate for the ISC lifetime is in the order of picosecundum (see Supplementary Note 9) which is much shorter than the radiative lifetime of the $^1E$ state. The $E_{x,y}$ state (or one of them in the presence of constant electric fields or strain, e.g., Ref. [35]) can be used as one of the quantum levels that can be initialized by optical pumping.

Here we note that the $^3E$ states are highly sensitive to strain or constant electric fields as manifested already by the strong electron-phonon coupling. Up to 0.64% strain could be applied to

monolayer $WS_2$[39]. As a comparison, 0.005% strain induces about 38 GHz energy splitting between $E_{x,y}$ in diamond NV centre[35]. Thus, the actual strain in $WS_2$ may largely split and shift the spin levels of $C_S^0$ that can be tuned towards the desired microwave resonance fields.

The optical lifetime of the $^3E$ state can be calculated with

$$\tau_{\rm rad} = p_1 \frac{3\pi\epsilon_0 c^3 \hbar^4}{n_D E_{\rm ZPL}^3 \mu^2}, \qquad (7)$$

where $\epsilon_0$ is the vacuum permittivity and $c$ is the speed of light. Here $n = 3.04$ is the refractive index of $WS_2$ and $\mu$ is the optical transition dipole moment. $p_1$ is the mix fraction of the singlet into $^3E$ state. Finally, the calculated optical lifetime is about 145 ns with a corresponding radiative rate at $1.10(2\pi)$ MHz (in angular frequency). We note that the non-radiative decay may compete with the radiative one at elevated temperatures which shorten the lifetime and decrease the quantum efficiency (see Supplementary Fig. 15). The estimated lifetime at low temperature implies that it is viable to apply quantum operation for this state, for instance, by employing a $\pi$ microwave pulse resonant with $A_2$ state. Since $A_{1,2}$ states are completely dark it should provide a large optical read-out contrast with respect to the $E_{x,y}$ states. Thus, the electron spin qubit state can be optically initialised and read out, as sketched in Fig. 4. According to recent studies[40], the coherence time of the $S = 1$ qubit should be about 11 ms in $WS_2$ thus the coherence time may be limited by the large hyperfine interaction of the electron to the nearest nuclear spins of the defect. According to our results (see Supplementary Note 7), $^{13}C$ has a large hyperfine interaction (in the order of 100 MHz), thus should be avoided as a potential ancilla qubit because it would cause decoherence. Although, the immediate three $^{183}W$ neighbour nuclear spins have an interaction strength of about 10 MHz which could also decoherent the electron spin. On the other hand, farther nuclear spins (see Supplementary Fig. 12) with an interaction strength of about 100 kHz may be used as a resource to store the quantum information before decaying to the ground state that may be realized at excited state level anticrossing with applying a small well tuned external magnetic field[41,42]. That qubit can be a very long living quantum memory in $WS_2$ host, similarly to the concept of ST1 centre in diamond[34]. We note that the ZPL emission could be small over the phonon-assisted emission because the estimated Debye-Waller factor is about 0.002 based on the Huang-Rhys theory applied to the $C_s$ geometry of the $^3E$ state (see Supplementary Note 6). We estimate about 1 million counts per second total emission from a single neutral C defect based on radiative lifetime and estimated collection

efficiency (19.1% in Ref. [43]) which can be readily observed. The coherent ZPL emission would be in the order of 2500 counts per second which is still observable. Nevertheless, optical cavities were already built from $WS_2$[44,45] that can be used to significantly enhance ZPL emission of $C_s$ qubit by Purcell effect.

In conclusion, we determined the electronic and vibronic properties of carbon defect in $WS_2$ that has been recently observed in experiments. The signal measured in STS could be attributed to the localized states in different charge states. The energy diagram of $C_S^0$ manifests itself as promising qubit candidate as the spin can be initialised and read out optically, and microwave excitation can be applied to prepare a coherent state. To produce the periodic pattern of this optically active defect, other unintentional impurities should be avoided surrounding the defects. A hBN capping layer can be used to prevent potential hydrogen reattachment (see Supplementary Note 2). Owing to the controllable fabrication, surface accessibility, mechanically and electrically tunable electro-optical properties, and optical cavities from the material, we propose $C_S$ in $WS_2$ as a scalable qubit operating at the near infrared region close to the ideal telecom wavelength.

## Methods

**Details on DFT calculations**. Our calculations are performed using the density functional theory (DFT) implemented in Vienna ab initio simulation package (VASP)[46,47]. Projector augmented wave (PAW) formalism[48,49] is selected to describe the valence electrons close to nuclei. The energy cutoff for the expansion of the plane-wave basis set was set to 520 eV. The screened hybrid density functional of Heyd, Scuseria, and Ernzerhof (HSE)[50,51] is used to calculate the electronic structure. In this approach, the short-range exchange potential is described by mixing with part of nonlocal Hartree-Fock exchange with fraction $\alpha$. The electron-electron interaction between short and long ranged part is adjusted by separation parameter $\mu$. To overcome this difficulty, these two parameters are adjusted. The HSE parameters were attested against many-body perturbation methods (see Supplementary Note 1), resulting in $\alpha = 0.40$ with $\mu = 0.10$ Å$^{-1}$, which could closely reproduce the experimental fundamental band gap at about 2.76 eV[32,52] with SOC interaction included as shown in Supplementary Fig. 2.

A $6 \times 6$ monolayer supercell is constructed and the perfect supercell contains 108 atoms which is sufficient to avoid the periodic defect-defect interaction, a vacuum layer of 12 Å is applied to separate the periodic layer images. We also consider a graphene substrate which is a direct simulation of the experiments,[32] together with calculation containing hBN capping layer to protect the defect, as shown in Supplementary Fig. 3. In this case we apply $4 \times 4$ supercell for $WS_2$ and $5 \times 5$ supercell for graphene or hBN that are nearly commensurate. The result shows the graphene or hBN substrate would not change the electronic structure of $WS_2$ and the defect levels are well kept. $\Gamma$-point scheme is converged for the k-point sampling of the Brillouin zone for these sizes of supercell. During structural optimization, the internal coordinates are allowed to relax until the forces of each atom are less than 0.01 eV/Å. The calculated lattice constant is 3.14 Å and agrees well with experimental value (3.155 Å)[53,54]. The optical transition energies were obtained from the differences in total energies between the different electronic configurations through $\Delta$SCF method which has a typical accuracy of about 0.1 eV with involving localized defect states[55]. We note that the Kohn-Sham orbitals change their localization in the $\Delta$SCF procedure with respect to those in the ground state electronic configuration for the neutral C defect. Our ab initio many-body perturbation theory based GW and Bethe-Salpeter equation (BSE) calculations partially support the construction of the lowest energy excitonic state of the neutral C defect in the $\Delta$SCF method as discussed in Supplementary Note 8. We note that $\Delta$SCF method for the negatively charged C defect may produce less accurate results than that for the neutral defect because the former involves delocalized states split from the conduction band that may be treated by many-body perturbation methods[56–58].

The phonon calculation at $\Gamma$-point scheme for the supercell is done by calculating Hessian matrix based on the derivative forces. The HSE optimized geometry is fixed to calculate the phonon at Perdew-Burke-Ernzerhof (PBE)[59] level. The SOC is calculated within the noncollinear approach. The spin quantization axis is aligned with symmetry axis of the defect. The SOC is evaluated based on geometry from spin-polarized calculation. Here supercell size scaling method, Freysoldt-Neugebauer-Van de Walle (FNV) correction and self-consistent charge correction[60,61] are used to calculate the formation energy and defect levels as discussed in Supplementary Note 3. The non-radiative transition is calculated based on NONRAD code[62], as discussed in Supplementary Note 9.

## Data availability

The authors declare that the main data supporting the findings of this study are available within the paper and its Supplementary files. Part of source data is provided with this

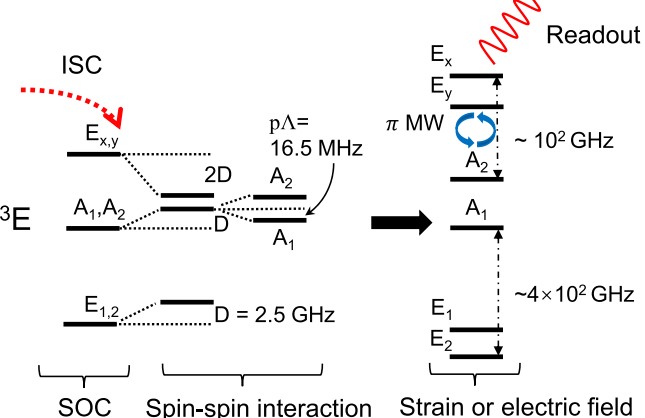

**Fig. 4 The spin-spin interaction and quantum protocol for $C_S^0$.** The splitting between $A_1$ and $A_2$ is reduced by Ham factor.

paper. Other data that support the findings of this study are available from the corresponding author upon reasonable request.

## Code availability

The codes that were used in this study are available upon request to the corresponding author.

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

## Acknowledgements

A.G. acknowledges the Hungarian NKFIH grant No. KKP129866 of the National Excellence Program of Quantum-coherent materials project and the support for the

Quantum Information National Laboratory from the Ministry of Innovation and Technology of Hungary, and the EU H2020 Quantum Technology Flagship project ASTERIQS (Grant No. 820394).

## Author contributions

S.L. carried out the DFT calculations under the supervision of A.G., P.U., G.T. and V.I. developed and applied the electron-phonon coupling theory on the defect under the supervision of A.G. All authors contributed to the discussion and writing the manuscript. A.G. conceived and led the entire scientific project.

## Funding

## Competing interests

The authors declare no competing interests.
