## [Peer Review File · Nature Communications]

REVIEWER COMMENTS

Reviewer #1 (Remarks to the Author):

This work by Li and coauthors seeks to understand the electronic structure and defect states of single carbon defect in monolayer WS₂. It is of great interest and important to identify novel qubits in 2D materials which could be controlled externally and produced deterministically in large scale. The authors were motivated by recent experiment work (arXiv:2008.12196) where a substitutional atomic defect (C_S) was deterministically created by scanning tunneling microscope. A series of electronic structure and total energy calculations were performed to extract defect formation energy and charge transition level, and to derive optical excitation cycles and understand the outcomes of the coupling among spin, orbital, and phonon at the defect states. Based on theoretical analysis and calculations, they proposed the triplet states of neutral C_S defect in WS₂ as a suitable qubit, and discussed a protocol to initialize and utilize the qubit. The results are novel and quite interesting which will motivate experimental validation and realization. I would recommend it for publication on Nature Communications if the following questions can be addressed properly.

* The authors pointed out that spin levels in WS₂ could be very sensitive to the strain. In real device, monolayer WS₂ with C_S defect may be subject to in plane strain due to the substrate effect. Additionally, to protect C_S defect, it may need to be encapsulated with additional top layer such as BN to prevent hydrogen reattachment. Will these external conditions severely affect final defect states and energy splitting and limit the application of this defect as qubit?

* Furthermore, because of different atomic registry between each C_S single atomic defect and top/bottom substrate, the deterministically created single C_S defects may differ from each other, preventing or hindering their use as a scalable qubit platform. These practical considerations need to be elaborated in detail.

* Vibrational modes coupled to defect electronic states could lead to spin decoherence. How could one balance this tradeoff for C_S in monolayer WS₂?

* On page 11, how was the optical lifetime (145 ns) determined?

* On page 4 of Supplementary Materials, it was claimed that "Without charge correction, the HSE calculated occupied state is 0.51 eV above VBM while the unoccupied state is above the conduction

band minimum (CBM)." But according to Figure 4b, without charge correction, the unoccupied state is below CBM, not above CBM.

* A minor typo in Eq. (2) of Supplementary Materials: C/a^3 should be D/a^3 . There are also some minor grammar errors, e.g. p8, "The eigenvalues of the matrix provide the electronic part of SOC are discussed in ...".

Reviewer #2 (Remarks to the Author):

Review

The manuscript by Li et al. discusses the electronic structure and properties of single carbon defects in TMDs. The authors use this case to suggest a theory-based protocol, associating atomic defects in TMDs with their potential to be used as qubits due to their charge and spin configurations. They apply density functional theory within the HSE approximation, for a supercell of monolayer WS₂ with Cs defect, and analyze the energy associated with the defect states upon defect charging. The authors then use their results to build a simplified picture of the singlet and triplet excited states, and couple them through an electron-phonon interaction Hamiltonian. The authors point to a telecom wavelength of the calculated ZPL energy and use the computed transition dipole moment to estimate radiative lifetimes and optical properties.

This study thus presents ab initio understanding of a specific type of defect, of potential interest for quantum information. The suggested computational approach is interesting, as it is relatively simple and accessible and yet seems reliable for the presented observables. However, I find that the main findings presented in this manuscript are methodological. The suggested qubit application of defect localized states and spin-orbit coupling is already widely discussed and well-studied for the specific case of atomic defects in TMDs. I think this work can and should be published in a specialized journal, but it lacks the innovation and the broad impact required for publication in Nature Communications.

Below are more specific comments and suggestions.

1. The exact-exchange parametrization applied within the HSE functional is tuned to optimize the bandgap compared to the experimental result. While this is fully explained in the SI, I think it should be also briefly discussed in the main manuscript. This is an important aspect of this study since usual HSE parametrization will not result in an experimental bandgap.

2. The energy levels associated with the defect are discussed and compared to experimental observations. The authors further discuss the change in formation energy and the change in state charging due to graphene substrate effects. For this case, I am not convinced that the HSE functional is good enough to describe the involved interactions. Since it does not include the correct $1/r$ long-range Coulomb decay (namely, only local exchange is considered in the long-range electron-electron interaction limit), this functional typically cannot properly describe interface effects. It is thus likely that the graphene effect shown here is limited, and the energy levels likely do not fully include interface screening effects. I believe this aspect should be further discussed in the manuscript.

3. Singlet and triplet configurations are described by the promotion of electrons to empty states, and analyzed by the spatial symmetry and spin configurations. However, TMDs host strongly bound excitons, and I am not sure if and how this is taken into account in the presented interaction picture. In addition, defect bands are well known to induce mixed excitonic states, with both defect-defect and defect-pristine transitions (see, e.g., E Mitterreiter et al., Nature Communications 12, 1-8, 2021). Such mixing is expected to change the spin configuration, and I believe it demands further discussion, even if not fully calculated.

4. It is not clear if the discussion of lifetime reduction is built on a calculation or a model, and how is it being performed. I believe a better explanation of the reported numbers- cooling of 100fs, slowing down by a factor of 50, optical lifetime of 145 ns – will better support the discussion.

Reviewer #3 (Remarks to the Author):

Comments to “Carbon defect qubit in two-dimensional WS₂” by Song Li et al

This work proposed to use C substitution of S defect in WS₂ as spin qubit and studied its electronic structure, electron-phonon coupling, spin-orbit coupling for the possibility of optical-readout. Most defect study in TMD has not been for spin related properties, rather on single photon emitters; therefore, this work has its importance from application point of view. Its proposal on optical readout from metastable state is also a good strategy for 2D materials where defects with triplet ground state are very rare. I suggest further proceed on publication after answering the following questions mostly regarding to technical points.

1. The authors picked C substitution of S at the beginning as the object of study, is there any reason for this choice? Is it more stable compared with other C defect configuration in this system? For example, interstitial, substitution of W, or C-S/W vacancies ?

2. This study considered monolayer TMD and the occupied bands are quite hybridized with VB in terms of energy range. In this situation, it is more difficult to neglect the mixed transition between defect and bulk states; what's more, single particle picture without considering excitonic effects is more difficult to justify compared to an isolated two-level system away from bulk states (as the excitonic effect will further mix the states in transition, while modify the excitation energy due to large exciton binding energy in 2D semiconductors). Can the authors make some comments on this point at least qualitatively?

3. An important point is to use metastable triplet state for optical readout. The authors mentioned the radiative lifetime is at ns, the ISC is at picosecond roughly. Can the authors give more details on how they are computed? Will these two rates be enough for assessment of readout efficiency? How about the competition with other nonradiative process since the electron-phonon coupling is quite strong in this system?

4. The HR factor is quite large and the phonon side band is also quite broad from SI. Can the authors comment on if this defect will be bright enough as SPE? Cavity is mentioned to improve the brightness; however it is always better to avoid further complexity if it's intrinsic or easily tuned.

REVIEWER COMMENTS

Reviewer #1 (Remarks to the Author):

This work by Li and coauthors seeks to understand the electronic structure and defect states of single carbon defect in monolayer WS₂. It is of great interest and important to identify novel qubits in 2D materials which could be controlled externally and produced deterministically in large scale. The authors were motivated by recent experiment work (arXiv:2008.12196) where a substitutional atomic defect (C_S) was deterministically created by scanning tunneling microscope. A series of electronic structure and total energy calculations were performed to extract defect formation energy and charge transition level, and to derive optical excitation cycles and understand the outcomes of the coupling among spin, orbital, and phonon at the defect states. Based on theoretical analysis and calculations, they proposed the triplet states of neutral C_S defect in WS₂ as a suitable qubit, and discussed a protocol to initialize and utilize the qubit. The results are novel and quite interesting which will motivate experimental validation and realization. I would recommend it for publication on Nature Communications if the following questions can be addressed properly.

We appreciate the overall positive feedback from the Reviewer. We read the comments and address them one by one. We hope that the improved manuscript will be persuasive in the eye of the readers.

* The authors pointed out that spin levels in WS₂ could be very sensitive to the strain. In real device, monolayer WS₂ with C_S defect may be subject to in plane strain due to the substrate effect. Additionally, to protect C_S defect, it may need to be encapsulated with additional top layer such as BN to prevent hydrogen reattachment. Will these external conditions severely affect final defect states and energy splitting and limit the application of this defect as qubit?

We thanks the Reviewer for highlighting issues about the possible challenges for which future experiments might face to. Indeed, strain induced by substrate due to the lattice mismatch is a very important aspect towards the electronic structure of the sample, for example the graphene on metal substrate. (*JOURNAL OF PHYSICS: CONDENSED MATTER* 27, 303002 (2015), *CARBON* 68, 440–451441 (2014)) To carefully evaluate the substrate effect, we had checked the original experimental paper together with similar studies (*PHYSICAL REVIEW B* 101, 081201(R) (2020), *ACS NANO* 13, 10520–10534 (2019), *NATURE COMMUNICATIONS* 10,3382 (2019), *PHYSICAL REVIEW LETTERS* 123, 076801 (2019), *2D MATERIALS* 5, 045010 (2018)). The WS₂ samples are grown using chemical vapor deposition (CVD) on graphitized SiC substrates and usually the WS₂ is supported by mixed single or bilayer graphene. Under this condition, the weak van der Waals (vdW) interaction between individual layers allows the growth of unstrained two-dimensional (2D) films that are crystallographically aligned to the underlying substrate even in the presence of a large lattice mismatch. Graphene provides a conductive support that is ideal for vertically integrated electronic devices and electronic spectroscopy such as scanning

tunneling spectroscopy (STS) or photoelectron spectroscopy. Previous STS spectra measured the sulfur vacancy in the top and bottom sulfur layer and found that they were electronically equivalent (*PHYSICAL REVIEW LETTERS* 123, 076801 (2019)). If the substrate could induce large influence on WS₂ sample, especially the strain effect, we believe the STS spectra would be different (bandgap decrease or energy shift for defect levels) for the two configurations. Therefore, the in-plane strain caused by the graphene substrate might be negligible.

Of course, we have to admit that the conclusion is not well guaranteed for other substrates or different experimental temperature. Recent strain engineering on 2D transition metal dichalcogenides (TMDC) is a promising field and the optical and electronic properties (photoluminescence, absorption) of TMDC can be precisely tuned by the lattice and thermal coefficient of expansion mismatch of substrate (*npj 2D MATERIALS AND APPLICATIONS* 10 (2017), *NATURE COMMUNICATIONS* 8, 608 (2017), *SCIENTIFIC REPORTS* 6, 35154 (2016)). As mentioned in the main text, this paves a way to researchers to design suitable WS₂ based quantum architectures with strain field.

Another important issue is the protection of the qubit from species in the environment. One possible route is to encapsulate the WS₂ layer into the sandwich of hexagonal boron nitride layers (hBN). Motivated by the Reviewer's comment, we carried out new calculations to test the influence of hBN capping layers. Similar to graphene substrate, we use 5 × 5 supercell of hBN layers on top and bottom and 4 × 4 supercell of WS₂ in the middle. The HSE band diagrams are plotted below.

Figure 1. The WS₂ with hBN substrate and capping layer and the band diagram.

We still use single Gamma point scheme to optimize the tri-layer structure considering that the HSE functional is time demanding. With the hBN included, the heterostructure shows type-I band alignment [in agreement with Ref. *PHYSICAL REVIEW B* 94, 035125 (2016)]. The defect level of the neutral charge state only shifts 0.06 eV upwards, and this is due to artificial strain in the system to match the lattice constants of the two materials. We emphasize that the commensurability is required by our simulation technique because of the periodic boundary condition but no such a constraint occurs in the experiment. The ions in a larger supercell consisting of 10 ×

10 hBN on top and 8×8 WS₂ were also optimized by the computationally less demanding PBE functional and then the electronic structure at the fixed coordinates was calculated by the accurate and expensive HSE functional. The defect levels are written in parentheses in Figure 1. The energy difference within the two systems is about 0.01 eV. The weak van der Waals interaction between the layers would not influence the electronic structure of WS₂ therefore hBN would be excellent protective layer. (*JOURNAL OF APPLIED PHYSICS* 122, 065303 (2017), *SURFACE SCIENCE* 672–673, 13–1814 (2018))

For the tri-layer structure, the ZPL of the triplet in neutral charge state is 0.889 eV (without the PJT⊗SOC) which is still close to the single layer result, 0.847 eV. Since the defect-defect interaction is larger in 4×4 supercell, the defect levels shift upward (0.6 eV), nevertheless, the localized wavefunction is still similar to the single layer case. This indicates that the calculated intersystem crossing (ISC) and non-radiative (NRD) rates (Q3 from reviewer 3) would not change significantly and do not alter our conclusions.

Figure 2. The occupied localized a_1 state in triplet state for neutral C_s in tri-layer and single layer model.

The description about the calculations on the protective hBN layers is added to Supplementary Note 3. And we mentioned this in Method part in main text “.....We also consider a graphene substrate which is a direct simulation of the experiments, together with calculation containing hBN capping layer to protect the defect, as shown in Supplementary Figure 3. In this case we apply 4×4 supercell for WS₂ and 5×5 supercell for graphene (hBN) that are nearly commensurate.....”

* Furthermore, because of different atomic registry between each C_S single atomic defect and top/bottom substrate, the deterministically created single C_S defects may differ from each other, preventing or hindering their use as a scalable qubit platform. These practical considerations need to be elaborated in detail.

We agree with the Reviewer’s comment. In experiment, the plasma-assisted strategy is used to incorporate carbon-hydrogen (CH) impurity and then STM tip can create single carbon substitution by hydrogen desorption. Whether the carbon radical ions (CRI) are charged or not, the UHV condition could stabilize the geometry and prevent other impurities from destroying them. We note that it is unlikely to have the CH impurity on the bottom layer of WS₂ due to the graphene/SiC substrate. Hence, the location and local environment of C_s defects ensemble are quite similar. We speculate protective capping layer like hexagonal boron nitride should not significantly change the scenario.

However, it is likely to produce sulfur vacancy in the sample, also this vacancy could be generated during hydrogen depassivation process. In addition, the removed hydrogen can transfer back and attach to carbon again. If the CRIs are close to these impurity defects, the electronic structure might be modified and degrades the quantum scalability. Therefore, to produce the periodic pattern of this optically active defect, the unintentional impurities should be avoided surrounding the defects.

Another way to create point defects in TMDCs is by focused He-ion irradiation with helium ion microscope. (*NATURE COMMUNICATIONS* 12, 3822 (2021), *NANO LETTERS*. 20, 4437–4444 (2020), *ACS PHOTONICS* 8, 669–677 (2021)) This might provide valuable information for our analysis about the possible defect types due to the lack of direct characterization of plasma-assisted CVD WS₂ sample. The incident helium ions sputter the sample and create different types of vacancies. Several kinds of dopants can be incorporated into vacancies like silicon, oxygen, and antisite dopants, etc. Although the density of these defects is lower than sulfur vacancy and the author indicate these defects do not impact macroscopic functionality, their influence on neighboring optical active defects is still elusive and need further study in the future.

We believe that all the scenarios sketched above cannot be the scope of a single paper. The encapsulation of the single layer WS₂ by hBN layers could solve many potential problems as discussed above. We add one sentence in the conclusion part to briefly discuss the practical consideration as you suggested.

Now it reads: "To produce the periodic pattern of this optically active defect, other unintentional impurities should be avoided surrounding the defects. A hBN capping layer can be used to prevent potential hydrogen reattachment."

* **Vibrational modes coupled to defect electronic states could lead to spin decoherence.**
How could one balance this tradeoff for C_S in monolayer WS₂?

The decoherence problem is associated with the large Huang-Rhys factor. As indicated in the main text, optical cavity is one efficient method to reduce the phonon side band and enhance the ZPL. This phenomenon has been demonstrated on NV center with cavity strongly coupled to NV ensemble in diamond. The decoherence can be suppressed and the amplitude of coherent oscillation between the cavity and spin ensemble can be enhanced by two orders of magnitude. (*NATURE PHYSICS* 10, 720–724 (2014)) We note, on the other hand, that coherent photons are only needed to generate a spin-to-photon interface for quantum communication. The entire emission including the emission from the phonon sideband can be used to read the spin state of the defect that can be useful in quantum sensing (similar to NV center in diamond).

* On page 11, how was the optical lifetime (145 ns) determined?

The radiative lifetime of the triplet can be calculated by:

$$\tau_{rad} = p \frac{3\pi\epsilon_0 c^3 \hbar^4}{n_D E_{ZPL}^3 \mu^2}$$

where ϵ_0 is the vacuum permittivity and c is the speed of light. Here $n_D = 3.04$ is the refractive index of WS₂ and μ is the optical transition dipole moment. p is the fraction of singlet character mixed with triplet, as radiative decay is expected only from the singlet component of the excited state towards the ground state singlet.

* On page 4 of Supplementary Materials, it was claimed that "Without charge correction, the HSE calculated occupied state is 0.51 eV above VBM while the unoccupied state is above the conduction band minimum (CBM)." But according to Figure 4b, without charge correction, the unoccupied state is below CBM, not above CBM.

In Fig. S4b, the KS level of the occupied and unoccupied states are extrapolated to infinite supercell size by PBE functionals since it is really time consuming to calculate this in a large supercell by HSE functionals. As shown in Fig. S3, we also did the calculation based on the PBE to get the band structure of WS₂ with graphene substrate and both the occupied and unoccupied defect states appear in the gap. PBE yields inaccurate result not only for the band gap but also for the defect levels. Here we would like to show the evolution trend of the defect levels with increasing the supercell size and the exact position of the defect levels should be considered with HSE functionals.

* A minor typo in Eq. (2) of Supplementary Materials: C/a^3 should be D/a^3 . There are also some minor grammar errors, e.g. p8, "The eigenvalues of the matrix provide the electronic part of SOC are discussed in ...".

Thank you for pointing these out. We modified the manuscript as you suggested.

For Eq. 2 in SM, C/a^3 changed to D/a^3 .

On p8, "The eigenvalues of the matrix including the electronic part and SOC are discussed in ..."

Reviewer #2 (Remarks to the Author):

Review

The manuscript by Li et al. discusses the electronic structure and properties of single carbon defects in TMDs. The authors use this case to suggest a theory-based protocol, associating atomic defects in TMDs with their potential to be used as qubits due to their charge and spin configurations. They apply density functional theory within the HSE approximation, for a supercell of monolayer WS₂ with Cs defect, and analyze the energy associated with the defect states upon defect charging. The authors then use their results to build a simplified picture of the singlet and triplet excited states, and couple them through an electron-phonon interaction Hamiltonian. The authors point to a telecom wavelength of the calculated ZPL energy and use the computed transition dipole moment to estimate radiative lifetimes and optical properties.

This study thus presents ab initio understanding of a specific type of defect, of potential interest for quantum information. The suggested computational approach is interesting, as it is relatively simple and accessible and yet seems reliable for the presented observables. However, I find that the main findings presented in this manuscript are methodological. The suggested qubit application of defect localized states and spin-orbit coupling is already widely discussed and well-studied for the specific case of atomic defects in TMDs. I think this work can and should be

published in a specialized journal, but it lacks the innovation and the broad impact required for publication in Nature Communications.

While we appreciate the Reviewer's effort and feedback, we respectfully disagree with certain points. In particular, our calculations are certainly not simple even though, the underlying theory is predominantly DFT throughout the paper. We wish to emphasize that ZPL energies can be only obtained by calculating the quantum mechanical forces in the excited states. In this procedure, we take into account the strong-electron phonon coupling and correlation between the electronic states. We apply the DFT based methods with great care with using the deep insight from the group theory analysis and intricate details of the many-body electron-ion system. To our knowledge, no protocol has been fully established to coherently manipulate and read out defect qubit spins in TMDs, in particular, for those ones which can be generated on demand at single defect level in experiment.

Below are more specific comments and suggestions.

1. The exact-exchange parametrization applied within the HSE functional is tuned to optimize the bandgap compared to the experimental result. While this is fully explained in the SI, I think it should be also briefly discussed in the main manuscript. This is an important aspect of this study since usual HSE parametrization will not result in an experimental bandgap.

Thanks for the kindly suggestion. We modified the main text according to the comments. We added sentences at the Method section and briefly discussed the corrected HSE functionals parameter we used here.

Now in Method part, it reads:

“.....The screened hybrid density functional of Heyd, Scuseria, and Ernzerhof (HSE) is used to calculate the electronic structure. In this approach, the short-range exchange potential is described by mixing with part of nonlocal Hartree–Fock exchange with fraction α . The electron-electron interaction between short and long ranged part is adjusted by separation parameter μ . The conventional HSE06 parameter can only yield optical band gap or smaller when SOC included. To overcome this difficulty, these two parameters are adjusted. The HSE parameters were attested against many-body perturbation methods (see Supplementary Note 1), resulting in $\alpha= 0.40$ with $\mu = 0.10 \text{ \AA}^{-1}$, which could closely reproduce the experimental fundamental band gap at about 2.76 eV with SOC interaction included as shown in Supplementary Figure 2.”

2. The energy levels associated with the defect are discussed and compared to experimental observations. The authors further discuss the change in formation energy and the change in state charging due to graphene substrate effects. For this case, I am not convinced that the HSE functional is good enough to describe the involved interactions. Since it does not include the correct $1/r$ long-range Coulomb decay (namely, only local exchange is considered in the long-range electron-electron interaction limit), this functional typically cannot properly describe interface effects. It is thus likely that the graphene effect shown here is limited, and the energy levels

likely do not fully include interface screening effects. I believe this aspect should be further discussed in the manuscript.

We agree that this could be a potential limitation of the study but we concluded that it is a minor effect for our special case.

Previous investigations reveal the metallic substrates dramatically renormalize the quasiparticle band gap compared to the free-standing one. (*NATURE MATERIALS* 13, 1091–1095 (2014), *ACS NANO* 10, 1067–1075 (2016), *NANO LETTERS* 14, 2443–2447 (2014), *PHYSICAL REVIEW MATERIALS* 2, 084002 (2018)) Here we would like to use MoS₂ as an example due to the lack of investigation on WS₂ about this issue. The band gap reduction can be calculated by mapping the in-plane separate MoS₂ irreducible polarizability to the substrate. For MoS₂/Gr, simulation shows the gap renormalization is about 350 meV and the error bar could be about 100 meV. (*PHYSICAL REVIEW MATERIALS* 2, 084002 (2018)) However, the experimental measured band gap of MoS₂/Gr varies from 1.9 to 2.4 eV (*ACS NANO* 10, 6315–6322 (2016), *NANO LETTERS* 14, 2443–2447 (2014), *ACS NANO* 10, 1067–1075 (2016)) therefore there is no well-established experimental data for comparison. The gap reduction is sensitive to environment like interlayer distance, the thickness of substrate, and the temperature, etc. In addition, the STM tip might also further modify the gap. Another work indicates that the substrate screening mainly reduces the exciton binding energy without significant change for optical gap. (*NATURE MATERIALS* 13, 1091–1095 (2014)) Based on these considerations, the concern about screening in TMDs is indeed legitimate.

In WS₂, the observed gap is about 2.7-2.8 eV with graphene substrate which is identical to the usual GW approximation result for pure WS₂ as we mentioned in the manuscript. It seems that the band reduction is not significant as it is in MoS₂/Gr. The GW calculated largest band gap with SOC included is roughly about 2.85 eV. (*PHYSICAL REVIEW B* 87, 155304 (2013)) These results imply that the band reduction should be less than 100 meV. In addition, the substrate screening also influences the charge transition level of the defect which has similar trend to the band evolution. Therefore, the basic conclusions of the paper do not alter.

3. Singlet and triplet configurations are described by the promotion of electrons to empty states, and analyzed by the spatial symmetry and spin configurations. However, TMDs host strongly bound excitons, and I am not sure if and how this is taken into account in the presented interaction picture. In addition, defect bands are well known to induce mixed excitonic states, with both defect-defect and defect-pristine transitions (see, e.g., E Mitterreiter et al., *Nature Communications* 12, 1-8, 2021). Such mixing is expected to change the spin configuration, and I believe it demands further discussion, even if not fully calculated.

Thanks for pointing this out. The excitonic effect is inherited in the Δ SCF method of the DFT functional, where the electron-hole wave function is created by constraint occupation of the Kohn-Sham wave functions. This procedure is valid when the nature of the excited state is well captured by the constraint occupation procedure. Δ SCF method obviously produces quantitatively good results if the excited state can be described as promoting an electron from one occupied defect level to a higher

(empty) defect level in the gap [e.g., see NV center in diamond, Phys. Rev. Lett. 103, 186404 (2009)]. It is well established that this method can work for such systems too in which the occupied defect level is resonant in the valence band in the electronic ground state configuration but pops up in the band gap when emptied (e.g., SiV center and related centers in diamond or W-center in silicon, *PHYSICAL REVIEW X* 8, 021063 (2018), *arXiv:2108.04283*). The second scenario occurs for the neutral C defect in WS₂ which is the central object in the present study. Furthermore, Δ SCF method provides quantum mechanical forces in the excited state. The ability of finding the global energy minimum atomic configuration in the excited state plays an essential role for understanding the true nature of the excited states because of the strong electron-phonon coupling. Without mapping the adiabatic potential energy surface in the excited state, the true fine structure of the excited state cannot be determined, e.g., Ham reduction factor in the spin-orbit splitting.

In the negative charge state of C defect in WS₂, the optical excited state includes resonance to the conduction band states. In this case, Δ SCF method may have shortcomings in describing the excited state properly where GW+BSE many-body perturbation method could be adequate as reported in the quoted paper and in a previous report [*PHYSICAL REVIEW LETTERS* 121, 167402 (2018)]. Since the excitonic wave function may be delocalized in this case, the optical transition probability may be weak. Nevertheless, we showed in our paper that the negatively charged C defect is not stable upon illumination and rather the neutral defect should be considered. The excited state of the negatively charged C-defect was only considered for completeness but Δ SCF method may produce there less accurate result than GW+BSE would do for vertical excitation. In experiment, however, the ZPL energy and the corresponding phonon sideband are observed, thus GW+BSE result cannot be directly compared to experimental data either.

We modified our manuscript in the conclusion part to give a comment about this point. Now it reads: "We note that Δ SCF method for the negatively charged C defect may produce less accurate results than that for the neutral defect because the former involves delocalized states split from the conduction band that may be treated by many-body perturbation methods [citation to the quoted paper above and the PRL paper]".

4. It is not clear if the discussion of lifetime reduction is built on a calculation or a model, and how is it being performed. I believe a better explanation of the reported numbers- cooling of 100fs, slowing down by a factor of 50, optical lifetime of 145 ns – will better support the discussion.

The radiative lifetime of the triplet can be calculated by:

$$\tau_{rad} = p \frac{3\pi\epsilon_0 c^3 \hbar^4}{n_D E_{ZPL}^3 \mu^2},$$

where ϵ_0 is the vacuum permittivity and c is the speed of light. Here $n_D = 3.04$ is the refractive index of WS₂ and μ is the optical transition dipole moment. p is the fraction of singlet character mixed into the triplet by spin-orbit interaction.

The ISC rate can be calculated by the application of first-order Fermi's golden rule,

$$\Gamma = \frac{2\pi}{\hbar} \lambda_z^2 F(\Delta E),$$

where ΔE is the energy spacing between singlet and triplet. F is the function of overlap between the vibrational spectrum of singlet and triplet. It can be approximated from the known phonon sideband in the PL spectrum within the HR approximation of the Franck-Condon theory of optical excitation. (*PHYSICAL REVIEW B* 91, 165201 (2015), *PHYSICAL REVIEW LETTERS* 114, 145502 (2015)). Since the singlet and triplet have the same electronic structure, only the axial component (λ_z) of SOC links the two states ($m_s = 0$).

We slightly modified the statement here about "slowing down by a factor of 50.....". The sentence now reads "slowed down for C_S^0 to about 38.2 ps because of the overlap spin function.....". The number here is the non-radiative decay (NRD) lifetime. This is now not an estimate from the nature of the state but we carried out new first principles calculations as explained in the response to Reviewer #3. The above two equations are added to the main text and the results about the temperature dependent ISC and NRD decay are described in Supplementary Note 8.

Reviewer #3 (Remarks to the Author):

Comments to "Carbon defect qubit in two-dimensional WS₂" by Song Li et al

This work proposed to use C substitution of S defect in WS₂ as spin qubit and studied its electronic structure, electron-phonon coupling, spin-orbit coupling for the possibility of optical-readout. Most defect study in TMD has not been for spin related properties, rather on single photon emitters; therefore, this work has its importance from application point of view. Its proposal on optical readout from metastable state is also a good strategy for 2D materials where defects with triplet ground state are very rare. I suggest further proceed on publication after answering the following questions mostly regarding to technical points.

We sincerely appreciate the valuable comments and suggestions from the Reviewer.

1. The authors picked C substitution of S at the beginning as the object of study, is there any reason for this choice? Is it more stable compared with other C defect configuration in this system? For example, interstitial, substitution of W, or C-S/W vacancies ?

The reason we choose the Cs defect is that it is experimentally fabricated by STM tips with atomic precision. Thus, there is no need a support from simulation to prove its existence. The Cs defect can be uniquely generated by hydrogen depassivation using STM tips for the carbon-hydrogen (CH) impurities which were induced by post-synthetic methane plasma treatment. This deterministic fabrication can create identical defects in a large scale which is crucial for solid state qubit platform.

It would be very helpful to systematically investigate other possible C related defects in WS₂ that might also present in differently treated WS₂. However, this issue is beyond the scope of the manuscript and we believe that exploring other defect

candidates or parasitic defects is for future work.

2. This study considered monolayer TMD and the occupied bands are quite hybridized with VB in terms of energy range. In this situation, it is more difficult to neglect the mixed transition between defect and bulk states; what's more, single particle picture without considering excitonic effects is more difficult to justify compared to an isolated two-level system away from bulk states (as the excitonic effect will further mix the states in transition, while modify the excitation energy due to large exciton binding energy in 2D semiconductors). Can the authors make some comments on this point at least qualitatively?

Thanks the Reviewer for pointing this out. In the constrained DFT method, the mixture of VB and defect level is considered and the wave function is optimized self-consistently. We found that the contribution from VB negligible since the self-consistent wave function of the hole becomes localized. We found very similar properties for other defects in diamond, silicon and silicon carbide (see the reply to Reviewer 2 that we do not reiterate here).

3. An important point is to use metastable triplet state for optical readout. The authors mentioned the radiative lifetime is at ns, the ISC is at picosecond roughly. Can the authors give more details on how they are computed? Will these two rates be enough for assessment of readout efficiency? How about the competition with other nonradiative process since the electron-phonon coupling is quite strong in this system? The radiative lifetime of the triplet can be calculated by

$$\tau_{rad} = p \frac{3\pi\epsilon_0 c^3 \hbar^4}{n_D E_{ZPL}^3 \mu^2},$$

where ϵ_0 is the vacuum permittivity and c is the speed of light. Here $n_D = 3.04$ is the refractive index of WS₂ and μ is the optical transition dipole moment. p is the fraction of singlet character mixed into the triplet state by the spin-orbit coupling.

The ISC rate can be calculated by the application of first-order Fermi's golden rule,

$$\Gamma = \frac{2\pi}{\hbar} \lambda_z^2 F(\Delta E),$$

where ΔE is the energy spacing between singlet and triplet. F is the function of overlap between the vibrational spectrum of the singlet and triplet states. It can be approximated from the known phonon sideband in the PL spectrum within the HR approximation of the Franck-Condon theory of optical excitation. (*PHYSICAL REVIEW B* 91, 165201 (2015), *PHYSICAL REVIEW LETTERS* 114, 145502 (2015)). Since the singlet and triplet wave functions have the same orbital part, only the axial component (λ_z) of SOC links the two states ($m_s = 0$).

To verify the efficiency of the optical protocol, we here further calculated the non-radiative decay rate from the excited state singlet to the triplet state and from the triplet state to the singlet ground state via following equations (*PHYSICAL REVIEW B* 90, 075202 (2014), *PHYSICAL REVIEW B* 100, 081407(R) (2019)):

$$\Gamma_{non} = \frac{2\pi}{\hbar} g W_{if}^2 X_{if}(T),$$

$$X_{if}(T) = \sum_{n,m} p_{in} |\langle \chi_i | \hat{Q} - Q_0 | \chi_f \rangle|^2 \times \delta(m\hbar\omega_i - m\hbar\omega_f + \Delta E_{if}),$$

$$W_{if} = \langle \psi_i | \partial_Q \hat{H} | \psi_f \rangle,$$

where W_{if} is the electronic term and $X_{if}(T)$ is phonon term with temperature dependency. g is the equivalent energy-degenerate atomic configurations and p_{in} is the thermal population. i and f correspond the initial and final state. The phonon matrix $|\langle \chi_i | \hat{Q} - Q_0 | \chi_f \rangle|$ is summed up the harmonic oscillator wave functions that enter the non-radiative recombination process. ΔE_{if} is the ZPL calculated in the main text and ψ is the single particle wavefunction from DFT calculation. The detailed calculation methodology (NONRAD) is given in Ref. *COMPUTER PHYSICS COMMUNICATIONS* 267, 108056 (2021).

We first calculate the configuration coordinate diagram which yields the phonon modes used in the following calculation.

Fig. 2 Configuration coordinate diagram of the ground (blue) and excited (orange) states.

Since the HR factor for singlet to triplet is 0.12, the lattice distortion from singlet to triplet is small and the calculated $dQ = 0.21 \text{ amu}^{1/2} \text{Å}$ while it is $1.72 \text{ amu}^{1/2} \text{Å}$ from triplet to ground state.

Fig. 3 Valence bands shift due to the el-ph coupling.

The electron-phonon coupling matrix can be evaluated by the shift of the occupied states. Here we select the top three valence bands. The calculated ISC and non-radiative rate are as following:

Fig. 4 ISC and non-radiative rates as a function of temperature from singlet 1E to triplet 3E states and triplet 3E state to ground state.

From 1E to 3E states, considering the large SOC and small energy splitting, the ISC rate at 1K is about 1.47 THz and the non-radiative decay (NRD) rate is 3.74 THz ($g = 3$ for this transition). Hence, the transition from the singlet 1E to triplet 3E excited states is very fast. The ISC and NRD rates between the triplet state and the ground state are very tiny (very slow processes) because the phonon overlap is extremely small due to the large change in the geometries. The conclusion is that the triplet state is indeed phosphorescent, predominantly via radiative emission because of the singlet mixture into the triplet state. The detailed discussion is added in Supplementary Note 8.

4. The HR factor is quite large and the phonon side band is also quite broad from SI. Can the authors comment on if this defect will be bright enough as SPE? Cavity is mentioned to improve the brightness; however it is always better to avoid further complexity if it's intrinsic or easily tuned.

We thank the Reviewer for the instructive comment. Now, we estimate the brightness of the neutral C defect with realistic experimental conditions. We find that by using present superconductor nanowire single photon detectors (SNSPD) the defect can be readily detected if the full emission of the defect is observed by the detector. The full emission can be used to read out the spin state as we stated in the response to Reviewer #1. To generate a spin-to-photon interface for quantum communication, only the coherent photons (ZPL emission) should be used which is about ~ 2500 counts/s that can be still observed by SNSPDs. The photon counts can be roughly estimated by the radiative lifetime (~ 6.9 MHz), quantum efficiency of the material ($\sim 19.1\%$) (*ADVANCED OPTICAL MATERIALS* 7, 1801270 (2018)), and the efficiency of detector which we assume 90% here. For applications, cavity is required to significantly enhance the coherent emission.

In the main text, we add the following in the discussion:

“...We estimate about 1 million counts per second emission from a single neutral C defect based on radiative lifetime and collection efficiency ($\sim 19.1\%$) (*ADVANCED OPTICAL MATERIALS* 7, 1801270 (2018)) which can be readily observed. However, the coherent emission would be in the order of 2500 counts per second which is still observable. Nevertheless, optical cavities were already built from WS_2 (*Nano Lett.* 16, 4368–4374 (2016) *Nano Lett.* 20, 3545–3552 (2020)) that can be used to significantly enhance ZPL emission by Purcell effect.”

REVIEWER COMMENTS

Reviewer #1 (Remarks to the Author):

The authors have addressed the comments and suggestions and revised the manuscript accordingly. I'd like to recommend it for publication on Nature Communications.

Reviewer #2 (Remarks to the Author):

The authors have properly addressed my comments, as well as comments from the other reviewers. The modified version includes a detailed explanation of the methods and models used. Still, my main concerns about the validity of their methods to exciton processes in TMDs still hold, as the use of Delta SCF methods does not account for electron-hole coupling to the best of my knowledge. I view this as the main limitation of the method presented in the manuscript, and I do not think it is properly discussed and analyzed in this work.

As I wrote in my previous report, I believe this manuscript is more suitable for a specialized issue, as, despite the methodological development, it lacks the broad impact required for publication in Nature Communications.

Reviewer #3 (Remarks to the Author):

The authors have sufficiently addressed my questions and I recommend it for publishing.

REVIEWER COMMENTS

Reviewer #2 (Remarks to the Author):

The authors have properly addressed my comments, as well as comments from the other reviewers. The modified version includes a detailed explanation of the methods and models used. Still, my main concerns about the validity of their methods to exciton processes in TMDs still hold, as the use of Delta SCF methods does not account for electron-hole coupling to the best of my knowledge. I view this as the main limitation of the method presented in the manuscript, and I do not think it is properly discussed and analyzed in this work.

As I wrote in my previous report, I believe this manuscript is more suitable for a specialized issue, as, despite the methodological development, it lacks the broad impact required for publication in Nature Communications.

We thank the Reviewer for agreeing most part of our modification of this manuscript.

We do not respectfully agree the Reviewer's comment that "Delta SCF method does not account for electron-hole coupling." In the Delta SCF method an electron is promoted from an occupied orbital (or band) to an unoccupied orbital (band). So, the electron-hole pair, which is the definition of an exciton quasi-particle, is constructed by the Delta SCF methodology. The electron-hole interaction in this picture is hidden in the self-consistent procedure of the DFT exchange-correlation calculation but naturally exists. On the other hand, the definition of "exciton binding energy" is not so clear. This issue is discussed in details in, e.g., Ref. [physica status solidi b, 248, 1337-1346 (2011)] where the time-dependent DFT (TDDFT) results were compared to the results from Delta SCF results on nitrogen-vacancy center in diamond. The exciton binding energy may be defined as the calculated Kohn-Sham gap between the occupied and unoccupied orbital in the ground state electronic configuration minus the calculated excitation energy at fixed geometry by the Delta SCF method (vertical excitation energy). Thus, Delta SCF method naturally and inherently contains the electron-hole coupling.

On the other hand, the construction of the exciton state is restricted to a single electron-hole pair in this procedure. The choice of the selection of the occupied and unoccupied (virtual) orbitals is based on the insight of the possible excitation routes but, in principle, it might be too restrictive and several pairs of occupied and virtual orbitals might be incorporated to describe the exciton wavefunction. In this regard, the Delta SCF method could be indeed criticized.

In other methods, such as GW+BSE method (Bethe-Salpeter equation), the construction of the exciton wavefunction is not restricted to a single electron-hole pair. However, the wavefunctions are fixed at the starting DFT (or DFT+scGW)

wavefunctions in the present implementations of BSE procedure. Therefore, BSE method gives a freedom in terms of the incorporated electron-hole pairs in the construction of the exciton but the result has no back action on the quasi-particle wavefunctions which may change the character of the quasi-particle wavefunctions building up the exciton itself.

In contrast, in Delta SCF method, often a single electron-hole pair is used to construct the exciton but the DFT exchange-correlation acts on the new set of Kohn-Sham orbitals and it can significantly modify its character with respect to the character in the ground state electronic configuration. As we noted in our reply in the previous round, the hole wavefunction is further localized and its level pops up in the band gap elevated from the valence band with respect to the ground state calculation. Thus, Delta SCF method gives a freedom for the back action on the Kohn-Sham wavefunctions building up the exciton. We believe that this is a highly critical issue for defects with resonant states in the bands which appeared in diamond, silicon carbide and silicon defects too with confirmed excitation energies within an accuracy of about 0.1 eV (see our published papers). Therefore, GW+BSE calculation cannot be used as a reference method for Delta SCF calculation for such electronic systems.

Nevertheless, we carried out new Delta SCF and GW+BSE calculations within Gamma-point approximation (see Fig. 1 below). We here use a smaller supercell, 4×4 , in order to reach close-to-converged calculation for the GW method which is very computationally demanding in the VASP implementation as more than 1500 bands were included in the single-shot G_0W_0 calculation. The energy cutoff for the response function is set to be 150 eV. The Tamm-Dancoff approximation was used to solve BSE. The highest seven valence bands and seven lowest conduction bands are considered as basis for the excitonic state in the BSE procedure. We found that the results do not change by including more valence bands in this procedure. The calculations are based on the optimized HSE functional which resulted in the optimized geometry and the electronic structure of the neutral carbon defect in WS_2 . Here, we do not consider spin-orbit interaction. The Delta SCF procedure is applied at fixed geometry as obtained in the ground state, so vertical excitation energy is calculated. Although, this procedure is not entirely converged because of the small supercell but provides qualitatively good BSE results.

We find that the empty a_1 defect level in the band gap by HSE agrees within 0.07 eV with G_0W_0 quasi-particle level with respect to the calculated valence band maximum. The Delta SCF method yields 1.43 eV vertical excitation energy in this supercell. The hole wavefunction is localized on the $e_{x,y}$ orbitals and the hole level pops up in the band gap, similarly to that in the large 6×6 supercell calculation. In the BSE absorption spectrum the first peak has a significant oscillator strength, and the weight of $e_{x,y}$ orbitals in the exciton state is about 86%. The dominant contribution of these orbitals in building up the hole part of the exciton justifies our procedure in Delta SCF. On the other hand, the first peak in the BSE absorption spectrum occurs at 1.77 eV. The source of

difference in the calculated Delta SCF and BSE excitation energies can be either the possibly missing 14% electron-hole pairs in building up the exciton in the Delta SCF procedure (with reducing the weight of the $e_{x,y}$ orbitals to 86%) or the missing back action on the quasi-particle wavefunction of the BSE method. According to our previous findings in other materials, the localization of the hole is highly critical which would decrease the excitation energy. Therefore, we conclude that the BSE method could justify the dominant contribution of the selected orbital in the Delta SCF method and the significant oscillator strength (optically allowed transition). On the other hand, the accuracy in the absolute value of the zero-phonon-line energy should be commented in the main text based on our previous findings.

To illustrate the importance of the change of the wavefunctions upon excitation, we describe the localization through estimating the number of atoms involved in the charge density which is the inverse participation ratio (IPR):

$$IPR = \frac{1}{\sum_i \rho_i^2},$$

where ρ_i is the site-projected charge density, and i is the atom site. The localization ratio β is defined as $\beta = NAT/IPR$, where NAT is the total number of atoms in supercell. Larger localization ratio indicates the more localized charge density or wavefunction. With small 4×4 supercell, after excitation, β of the empty e state is 4.94 which is relatively more localized than the occupied one with $\beta = 4.34$. For a_1 state, it becomes less localized going from the ground state (4.71) to excited state (4.42) electronic configuration. The overlap between the electron and hole parts of the exciton wavefunction increases which should result in a larger attractive electron-hole interaction making the excitation energy lower with respect to the wavefunctions taken from the ground state electronic configurations. We also calculate the localization parameters in the convergent 6×6 supercell where β of the empty e state is 7.35 and the occupied one with $\beta = 4.87$. β of the a_1 state is 10.44 at the ground state and it is 8.69 after excitation. During the excitation process, the e_x orbital (hole) is more localized while the a_1 (electron) orbital is less localized which increases the overlap between the densities of the two.

Orbital	4×4			6×6		
	e_x	e_y	a_1	e_x	e_y	a_1
GS	4.44, 1	4.44, 1	4.71, 0	4.72, 1	4.72, 1	10.44, 0
EX	4.94, 0	4.34, 1	4.42, 1	7.35, 0	4.87, 1	8.69, 1

Table 1. The localization ratio β and occupation numbers for defect levels at ground state and excited state electronic configurations as calculated by DFT and Delta SCF method within DFT, respectively.

Figure 1. GW+BSE calculation for 4×4 WS₂ supercell. (a) G_0W_0 band diagram of neutral carbon defect in WS₂. Numbers with red color indicate the calculated energy with HSE functionals. (b) the first two are HSE wavefunction of the degenerate e state and the third is the Δ SCF wavefunction and then the exciton wavefunction built upon the linear combination of states close to VB based on the BSE transition coefficient. The isosurface is 0.002 e/Å³. (c) The BSE fat band for the first and second BSE eigenstates at Γ point. The e to a_1 contribute the most to BSE eigenstate. (d) The optical transition intensity calculated with BSE. The first two peaks correspond to the e to a_1 transition.